# Multi-relational Poincaré Graph Embeddings

Ivana Balažević[1]     Carl Allen[1]     Timothy Hospedales[1,2]

[1] School of Informatics, University of Edinburgh, UK
[2] Samsung AI Centre, Cambridge, UK
{ivana.balazevic, carl.allen, t.hospedales}@ed.ac.uk

## Abstract

Hyperbolic embeddings have recently gained attention in machine learning due to their ability to represent hierarchical data more accurately and succinctly than their Euclidean analogues. However, multi-relational knowledge graphs often exhibit multiple simultaneous hierarchies, which current hyperbolic models do not capture. To address this, we propose a model that embeds multi-relational graph data in the Poincaré ball model of hyperbolic space. Our Multi-Relational Poincaré model (MuRP) learns relation-specific parameters to transform entity embeddings by Möbius matrix-vector multiplication and Möbius addition. Experiments on the hierarchical WN18RR knowledge graph show that our Poincaré embeddings outperform their Euclidean counterpart and existing embedding methods on the link prediction task, particularly at lower dimensionality.

## 1 Introduction

Hyperbolic space can be thought of as a continuous analogue of discrete trees, making it suitable for modelling *hierarchical* data [28, 10]. Various types of hierarchical data have recently been embedded in hyperbolic space [25, 26, 16, 32], requiring relatively few dimensions and achieving promising results on downstream tasks. This demonstrates the advantage of modelling tree-like structures in spaces with constant negative curvature (hyperbolic) over zero-curvature spaces (Euclidean).

Certain data structures, such as knowledge graphs, often exhibit multiple hierarchies simultaneously. For example, *lion* is near the top of the animal food chain but near the bottom in a tree of taxonomic mammal types [22]. Despite the widespread use of hyperbolic geometry in representation learning, the only existing approach to embedding *hierarchical multi-relational graph data* in hyperbolic space [31] does not outperform Euclidean models. The difficulty with representing multi-relational data in hyperbolic space lies in finding a way to represent entities (nodes), shared across relations, such that they form a different hierarchy under different relations, e.g. nodes near the root of the tree under one relation may be leaf nodes under another. Further, many state-of-the-art approaches to modelling multi-relational data, such as DistMult [37], ComplEx [34], and TuckER [2] (i.e. *bilinear models*), rely on *inner product* as a similarity measure and there is no clear correspondence to the Euclidean inner product in hyperbolic space [32] by which these models can be converted. Existing translational approaches that use *Euclidean distance* to measure similarity, such as TransE [6] and STransE [23], can be converted to the hyperbolic domain, but do not currently compete with the bilinear models in terms of predictive performance. However, it has recently been shown in the closely related field of word embeddings [1] that the difference (i.e. relation) between word pairs that form analogies manifests as a *vector offset*, suggesting a translational approach to modelling relations.

In this paper, we propose MuRP, a theoretically inspired method to embed hierarchical multi-relational data in the *Poincaré ball* model of hyperbolic space. By considering the surface area of a hypersphere of increasing radius centered at a particular point, Euclidean space can be seen to "grow" polynomially, whereas in hyperbolic space the equivalent growth is exponential [10]. Therefore, moving outwards from the root of a tree, there is more "room" to separate leaf nodes in hyperbolic space than in

Euclidean. MuRP learns relation-specific parameters that transform entity embeddings by *Möbius matrix-vector multiplication* and *Möbius addition* [35]. The model outperforms not only its Euclidean counterpart, but also current state-of-the-art models on the link prediction task on the *hierarchical* WN18RR dataset. We also show that our Poincaré embeddings require far fewer dimensions than Euclidean embeddings to achieve comparable performance. We visualize the learned embeddings and analyze the properties of the Poincaré model compared to its Euclidean analogue, such as convergence rate, performance per relation, and influence of embedding dimensionality.

## 2 Background and preliminaries

**Multi-relational link prediction** A *knowledge graph* is a multi-relational graph representation of a collection $\mathcal{F}$ of *facts* in triple form $(e_s, r, e_o) \in \mathcal{E} \times \mathcal{R} \times \mathcal{E}$, where $\mathcal{E}$ is the set of entities (nodes) and $\mathcal{R}$ is the set of binary relations (typed directed edges) between them. If $(e_s, r, e_o) \in \mathcal{F}$, then subject entity $e_s$ is related to object entity $e_o$ by relation $r$. Knowledge graphs are often incomplete, so the aim of *link prediction* is to infer other true facts. Typically, a *score function* $\phi : \mathcal{E} \times \mathcal{R} \times \mathcal{E} \to \mathbb{R}$ is learned, that assigns a score $s = \phi(e_s, r, e_o)$ to each triple, indicating the strength of prediction that a particular triple corresponds to a true fact. A non-linearity, such as the logistic sigmoid function, is often used to convert the score to a predicted probability $p = \sigma(s) \in [0, 1]$ of the triple being true.

Knowledge graph relations exhibit multiple properties, such as symmetry, asymmetry, and transitivity. Certain knowledge graph relations, such as *hypernym* and *has_part*, induce a *hierarchical* structure over entities, suggesting that embedding them in *hyperbolic* rather than Euclidean space may lead to improved representations [28, 25, 26, 14, 32]. Based on this intuition, we focus on embedding multi-relational knowledge graph data in hyperbolic space.

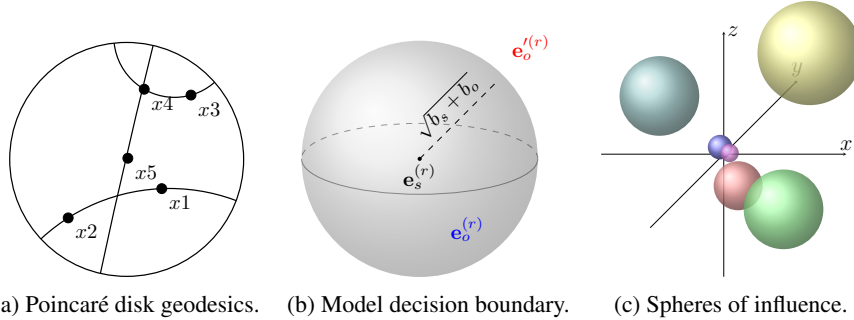

(a) Poincaré disk geodesics.　(b) Model decision boundary.　(c) Spheres of influence.

Figure 1: (a) Geodesics in the Poincaré disk, indicating the shortest paths between pairs of points. (b) The model predicts the triple $(e_s, r, e_o)$ as true and $(e_s, r, e_o')$ as false. (c) Each entity embedding has a *sphere of influence*, whose radius is determined by the entity-specific bias.

**Hyperbolic geometry of the Poincaré ball** The Poincaré ball $(\mathbb{B}_c^d, g^{\mathbb{B}})$ of radius $1/\sqrt{c}, c > 0$ is a $d$-dimensional manifold $\mathbb{B}_c^d = \{\mathbf{x} \in \mathbb{R}^d : c\|\mathbf{x}\|^2 < 1\}$ equipped with the Riemannian metric $g^{\mathbb{B}}$ which is *conformal* to the Euclidean metric $g^{\mathbb{E}} = \mathbf{I}_d$ with the *conformal factor* $\lambda_{\mathbf{x}}^c = 2/(1 - c\|\mathbf{x}\|^2)$, i.e. $g^{\mathbb{B}} = (\lambda_{\mathbf{x}}^c)^2 g^{\mathbb{E}}$. The distance between two points $\mathbf{x}, \mathbf{y} \in \mathbb{B}_c^d$ is measured along a *geodesic* (i.e. shortest path between the points, see Figure 1a) and is given by:

$$d_{\mathbb{B}}(\mathbf{x}, \mathbf{y}) = \frac{2}{\sqrt{c}} \tanh^{-1}(\sqrt{c}\| - \mathbf{x} \oplus_c \mathbf{y}\|), \tag{1}$$

where $\| \cdot \|$ denotes the Euclidean norm and $\oplus_c$ represents *Möbius addition* [35]:

$$\mathbf{x} \oplus_c \mathbf{y} = \frac{(1 + 2c\langle \mathbf{x}, \mathbf{y}\rangle + c\|\mathbf{y}\|^2)\mathbf{x} + (1 - c\|\mathbf{x}\|^2)\mathbf{y}}{1 + 2c\langle \mathbf{x}, \mathbf{y}\rangle + c^2\|\mathbf{x}\|^2\|\mathbf{y}\|^2}, \tag{2}$$

with $\langle \cdot, \cdot \rangle$ being the Euclidean inner product. Ganea et al. [13] show that *Möbius matrix-vector multiplication* can be obtained by projecting a point $\mathbf{x} \in \mathbb{B}_c^d$ onto the tangent space at $\mathbf{0} \in \mathbb{B}_c^d$ with the *logarithmic map* $\log_{\mathbf{0}}^c(\mathbf{x})$, performing matrix multiplication by $\mathbf{M} \in \mathbb{R}^{d \times k}$ in the Euclidean tangent space, and projecting back to $\mathbb{B}_c^d$ via the *exponential map* at $\mathbf{0}$, i.e.:

$$\mathbf{M} \otimes_c \mathbf{x} = \exp_{\mathbf{0}}^c(\mathbf{M}\log_{\mathbf{0}}^c(\mathbf{x})). \tag{3}$$

For the definitions of exponential and logarithmic maps, see Appendix A.

## 3 Related work

### 3.1 Hyperbolic geometry

Embedding hierarchical data in hyperbolic space has recently gained popularity in representation learning. Nickel and Kiela [25] first embedded the *transitive closure*[1] of the WordNet noun hierarchy, in the Poincaré ball, showing that low-dimensional hyperbolic embeddings can significantly outperform higher-dimensional Euclidean embeddings in terms of both representation capacity and generalization ability. The same authors subsequently embedded hierarchical data in the Lorentz model of hyperbolic geometry [26].

Ganea et al. [13] introduced Hyperbolic Neural Networks, connecting hyperbolic geometry with deep learning. They build on the definitions for Möbius addition, Möbius scalar multiplication, exponential and logarithmic maps of Ungar [35] to derive expressions for linear layers, bias translation and application of non-linearity in the Poincaré ball. Hyperbolic analogues of several other algorithms have been developed since, such as Poincaré GloVe [32] and Hyperbolic Attention Networks [16]. More recently, Gu et al. [15] note that data can be non-uniformly hierarchical and learn embeddings on a *product manifold* with components of different curvature: spherical, hyperbolic and Euclidean. To our knowledge, only Riemannian TransE [31] seeks to embed *multi-relational* data in hyperbolic space, but the Riemannian translation method fails to outperform Euclidean baselines.

### 3.2 Link prediction for knowledge graphs

**Bilinear models** typically represent relations as linear transformations acting on entity vectors. An early model, RESCAL [24], optimizes a score function $\phi(e_s, r, e_o) = \mathbf{e}_s^\top \mathbf{M}_r \mathbf{e}_o$, containing the bilinear product between the subject entity embedding $\mathbf{e}_s$, a full rank relation matrix $\mathbf{M}_r$ and the object entity embedding $\mathbf{e}_o$. RESCAL is prone to overfitting due to the number of parameters per relation being quadratic relative to the number per entity. DistMult [37] is a special case of RESCAL with diagonal relation matrices, reducing parameters per relation and controlling overfitting. However, due to its symmetry, DistMult cannot model asymmetric relations. ComplEx [34] extends DistMult to the complex domain, enabling asymmetry to be modelled. TuckER [2] performs a Tucker decomposition of the tensor of triples, which enables multi-task learning between different relations via the core tensor. The authors show each of the linear models above to be a special case of TuckER.

**Translational models** regard a relation as a translation (or vector offset) from the subject to the object entity embeddings. These models include TransE [6] and its many successors, e.g. FTransE [12], STransE [23]. The score function for translational models typically considers Euclidean distance between the translated subject entity embedding and the object entity embedding.

## 4 Multi-relational Poincaré embeddings

A set of entities can form different hierarchies under different relations. In the WordNet knowledge graph [22], the *hypernym*, *has_part* and *member_meronym* relations each induce different hierarchies over the same set of entities. For example, the noun *chair* is a parent node to different chair types (e.g. *folding_chair*, *armchair*) under the relation *hypernym* and both *chair* and its types are parent nodes to parts of a typical chair (e.g. *backrest*, *leg*) under the relation *has_part*. An ideal embedding model should capture all hierarchies simultaneously.

**Score function** As mentioned above, bilinear models measure similarity between the subject entity embedding (after relation-specific transformation) and an object entity embedding using the Euclidean inner product [24, 37, 34, 2]. However, a clear correspondence to the Euclidean inner product does not exist in hyperbolic space [32]. The Euclidean inner product can be expressed as a function of Euclidean distance and norms, i.e. $\langle \mathbf{x}, \mathbf{y} \rangle = \frac{1}{2}(-d_\mathbb{E}(\mathbf{x}, \mathbf{y})^2 + \|\mathbf{x}\|^2 + \|\mathbf{y}\|^2)$, $d_\mathbb{E}(\mathbf{x}, \mathbf{y}) = \|\mathbf{x} - \mathbf{y}\|$. Noting this, in Poincaré GloVe, Tifrea et al. [32] absorb squared norms into biases $b_\mathbf{x}, b_\mathbf{y}$ and replace the Euclidean with the Poincaré distance $d_\mathbb{B}(\mathbf{x}, \mathbf{y})$ to obtain the hyperbolic version of GloVe [27].

Separately, it has recently been shown in the closely related field of word embeddings that statistics pertaining to *analogies* naturally contain linear structures [1], explaining why similar linear structure

appears amongst word embeddings of word2vec [20, 21, 19]. Analogies are word relationships of the form "$w_a$ is to $w_a^*$ as $w_b$ is to $w_b^*$", such as "*man* is to *woman* as *king* is to *queen*", and are in principle not restricted to two pairs (e.g. "...as *brother* is to *sister*"). It can be seen that analogies have much in common with *relations* in multi-relational graphs, as a difference between pairs of words (or entities) common to all pairs, e.g. if $(e_s, r, e_o)$ and $(e_s', r, e_o')$ hold, then we could say "$e_s$ is to $e_o$ as $e_s'$ is to $e_o'$". Of particular relevance is the demonstration that the common difference, i.e. relation, between the word pairs (e.g. (*man*, *woman*) and (*king*, *queen*)) manifests as a *common vector offset* [1], justifying the previously heuristic translational approach to modelling relations.

Inspired by these two ideas, we define the basis score function for multi-relational graph embedding:

$$\phi(e_s, r, e_o) = -d(\mathbf{e}_s^{(r)}, \mathbf{e}_o^{(r)})^2 + b_s + b_o$$
$$= -d(\mathbf{R}\mathbf{e}_s, \mathbf{e}_o + \mathbf{r})^2 + b_s + b_o, \tag{4}$$

where $d : \mathcal{E} \times \mathcal{R} \times \mathcal{E} \to \mathbb{R}^+$ is a distance function, $\mathbf{e}_s, \mathbf{e}_o \in \mathbb{R}^d$ are the embeddings and $b_s, b_o \in \mathbb{R}$ scalar biases of the subject and object entities $e_s$ and $e_o$ respectively. $\mathbf{R} \in \mathbb{R}^{d \times d}$ is a diagonal relation matrix and $\mathbf{r} \in \mathbb{R}^d$ a translation vector (i.e. vector offset) of relation $r$. $\mathbf{e}_s^{(r)} = \mathbf{R}\mathbf{e}_s$ and $\mathbf{e}_o^{(r)} = \mathbf{e}_o + \mathbf{r}$ represent the subject and object entity embeddings after applying the respective relation-specific transformations, a stretch by $\mathbf{R}$ to $\mathbf{e}_s$ and a translation by $\mathbf{r}$ to $\mathbf{e}_o$.

**Hyperbolic model** Taking the hyperbolic analogue of Equation 4, we define the score function for our *Multi-Relational Poincaré (MuRP)* model as:

$$\phi_{\text{MuRP}}(e_s, r, e_o) = -d_{\mathbb{B}}(\mathbf{h}_s^{(r)}, \mathbf{h}_o^{(r)})^2 + b_s + b_o$$
$$= -d_{\mathbb{B}}(\exp_\mathbf{0}^c(\mathbf{R}\log_\mathbf{0}^c(\mathbf{h}_s)), \mathbf{h}_o \oplus_c \mathbf{r}_h)^2 + b_s + b_o, \tag{5}$$

where $\mathbf{h}_s, \mathbf{h}_o \in \mathbb{B}_c^d$ are *hyperbolic embeddings* of the subject and object entities $e_s$ and $e_o$ respectively, and $\mathbf{r}_h \in \mathbb{B}_c^d$ is a *hyperbolic translation vector* of relation $r$. The relation-adjusted subject entity embedding $\mathbf{h}_s^{(r)} \in \mathbb{B}_c^d$ is obtained by *Möbius matrix-vector multiplication*: the original subject entity embedding $\mathbf{h}_s \in \mathbb{B}_c^d$ is projected to the tangent space of the Poincaré ball at $\mathbf{0}$ with $\log_\mathbf{0}^c$, transformed by the diagonal relation matrix $\mathbf{R} \in \mathbb{R}^{d \times d}$, and then projected back to the Poincaré ball by $\exp_\mathbf{0}^c$. The relation-adjusted object entity embedding $\mathbf{h}_o^{(r)} \in \mathbb{B}_c^d$ is obtained by *Möbius addition* of the relation vector $\mathbf{r}_h \in \mathbb{B}_c^d$ to the object entity embedding $\mathbf{h}_o \in \mathbb{B}_c^d$. Since the relation matrix $\mathbf{R}$ is diagonal, the number of parameters of MuRP increases *linearly* with the number of entities and relations, making it scalable to large knowledge graphs. To obtain the predicted probability of a fact being true, we apply the logistic sigmoid to the score, i.e. $\sigma(\phi_{\text{MuRP}}(e_s, r, e_o))$.

To directly compare the properties of hyperbolic embeddings with the Euclidean, we implement the Euclidean version of Equation 4 with $d(\mathbf{e}_s^{(r)}, \mathbf{e}_o^{(r)}) = d_{\mathbb{E}}(\mathbf{e}_s^{(r)}, \mathbf{e}_o^{(r)})$. We refer to this model as the *Multi-Relational Euclidean (MuRE)* model.

**Geometric intuition** We see from Equation 4 that the biases $b_s, b_o$ determine the radius of a *hypersphere decision boundary* centered at $\mathbf{e}_s^{(r)}$. Entities $e_s$ and $e_o$ are predicted to be related by $r$ if relation-adjusted $\mathbf{e}_o^{(r)}$ falls within a hypershpere of radius $\sqrt{b_s + b_o}$ (see Figure 1b). Since biases are subject and object entity-specific, each subject-object pair induces a different decision boundary. The relation-specific parameters $\mathbf{R}$ and $\mathbf{r}$ determine the position of the relation-adjusted embeddings, but the radius of the entity-specific decision boundary is independent of the relation. The score function in Equation 4 resembles the score functions of existing translational models [6, 12, 23], with the main difference being the entity-specific biases, which can be seen to change the geometry of the model. Rather than considering an entity as a point in space, each bias defines an entity-specific *sphere of influence* surrounding the center given by the embedding vector (see Figure 1c). The overlap between spheres measures relatedness between entities. We can thus think of each relation as moving the spheres of influence in space, so that only the spheres of subject and object entities that are connected under that relation overlap.

## 4.1 Training and Riemannian optimization

We use the standard data augmentation technique [11, 18, 2] of adding reciprocal relations for every triple, i.e. we add $(e_o, r^{-1}, e_s)$ for every $(e_s, r, e_o)$. To train both models, we generate $k$ negative samples for each true triple $(e_s, r, e_o)$, where we corrupt either the object $(e_s, r, e_o')$ or the subject

$(e_o, r^{-1}, e'_s)$ entity with a randomly chosen entity from the set of all entities $\mathcal{E}$. Both models are trained to minimize the Bernoulli negative log-likelihood loss:

$$\mathcal{L}(y, p) = -\frac{1}{N} \sum_{i=1}^{N} (y^{(i)} \log(p^{(i)}) + (1 - y^{(i)}) \log(1 - p^{(i)})), \tag{6}$$

where $p$ is the predicted probability, $y$ is the binary label indicating whether a sample is positive or negative and $N$ is the number of training samples.

For fairness of comparison, we optimize the Euclidean model using stochastic gradient descent (SGD) and the hyperbolic model using *Riemannian stochastic gradient descent (RSGD)* [5]. We note that the Riemannian equivalent of adaptive optimization methods has recently been developed [3], but leave replacing SGD and RSGD with their adaptive equivalent to future work. To compute the Riemannian gradient $\nabla_R \mathcal{L}$, the Euclidean gradient $\nabla_E \mathcal{L}$ is multiplied by the inverse of the Poincaré metric tensor, i.e. $\nabla_R \mathcal{L} = 1/(\lambda_\theta^c)^2 \nabla_E \mathcal{L}$. Instead of the Euclidean update step $\theta \leftarrow \theta - \eta \nabla_E \mathcal{L}$, a first order approximation of the true Riemannian update, we use $\exp_\theta^c$ to project the gradient $\nabla_R \mathcal{L} \in T_\theta \mathbb{B}_c^d$ onto its corresponding geodesic on the Poincaré ball and compute the Riemannian update $\theta \leftarrow \exp_\theta^c(-\eta \nabla_R \mathcal{L})$, where $\eta$ denotes the learning rate.

## 5   Experiments

To evaluate both Poincaré and Euclidean models, we first test their performance on the knowledge graph link prediction task using standard WN18RR and FB15k-237 datasets:

**FB15k-237** [33] is a subset of Freebase [4], a collection of real world facts, created from FB15k [6] by removing the inverse of many relations from validation and test sets to make the dataset more challenging. FB15k-237 contains 14,541 entities and 237 relations.

**WN18RR** [11] is a subset of WordNet [22], a hierarchical collection of relations between words, created in the same way as FB15k-237 from WN18 [6], containing 40,943 entities and 11 relations.

To demonstrate the usefulness of MuRP on hierarchical datasets (given WN18RR is hierarchical and FB15k-237 is not, see Section 5.3), we also perform experiments on NELL-995 [36], containing 75,492 entities and 200 relations, $\sim 22\%$ of which hierarchical. We create several subsets of the original dataset by varying the proportion of non-hierarchical relations, as described in Appendix B.

We evaluate each triple from the test set by generating $n_e$ (where $n_e$ denotes number of entities in the dataset) *evaluation triples*, which are created by combining the test entity-relation pair with all possible entities $\mathcal{E}$. The scores obtained for each evaluation triple are ranked. All true triples are removed from the evaluation triples apart from the current test triple, i.e. the commonly used *filtered setting* [6]. We evaluate our models using the evaluation metrics standard across the link prediction literature: mean reciprocal rank (MRR) and hits@$k$, $k \in \{1, 3, 10\}$. Mean reciprocal rank is the average of the inverse of a mean rank assigned to the true triple over all $n_e$ evaluation triples. Hits@$k$ measures the percentage of times the true triple appears in the top $k$ ranked evaluation triples.

### 5.1   Implementation details

We implement both models in PyTorch and make our code, as well as all the subsets of the NELL-995 dataset, publicly available.[2] We choose the learning rate from $\{1, 5, 10, 20, 50, 100\}$ by MRR on the validation set and find that the best learning rate is 50 for WN18RR and 10 for FB15k-237 for both models. We initialize all embeddings near the origin where distances are small in hyperbolic space, similar to [25]. We set the batch size to 128 and the number of negative samples to 50. In all experiments, we set the curvature of MuRP to $c = 1$, since preliminary experiments showed that any material change reduced performance.

### 5.2   Link prediction results

Table 1 shows the results obtained for both datasets. As expected, MuRE performs slightly better on the non-hierarchical FB15k-237 dataset, whereas MuRP outperforms on WN18RR which contains

Table 1: Link prediction results on WN18RR and FB15k-237. Best results in bold and underlined, second best in bold. The RotatE [30] results are reported without their self-adversarial negative sampling (see Appendix H in the original paper) for fair comparison.

|  | WN18RR | | | | FB15k-237 | | | |
|---|---|---|---|---|---|---|---|---|
|  | MRR | Hits@10 | Hits@3 | Hits@1 | MRR | Hits@10 | Hits@3 | Hits@1 |
| TransE [6] | .226 | .501 | – | – | .294 | .465 | – | – |
| DistMult [37] | .430 | .490 | .440 | .390 | .241 | .419 | .263 | .155 |
| ComplEx [34] | .440 | .510 | .460 | .410 | .247 | .428 | .275 | .158 |
| Neural LP [38] | – | – | – | – | .250 | .408 | – | – |
| MINERVA [9] | – | – | – | – | – | .456 | – | – |
| ConvE [11] | .430 | .520 | .440 | .400 | .325 | .501 | .356 | .237 |
| M-Walk [29] | .437 | – | .445 | .414 | – | – | – | – |
| TuckER [2] | .470 | .526 | .482 | **.443** | **.358** | **.544** | **.394** | **.266** |
| RotatE [30] | – | – | – | – | .297 | .480 | .328 | .205 |
| MuRE $d = 40$ | .459 | .528 | .474 | .429 | .315 | .493 | .346 | .227 |
| MuRE $d = 200$ | .475 | .554 | .487 | .436 | **.336** | **.521** | **.370** | **.245** |
| MuRP $d = 40$ | **.477** | **.555** | **.489** | .438 | .324 | .506 | .356 | .235 |
| MuRP $d = 200$ | **.481** | **.566** | **.495** | **.440** | .335 | .518 | .367 | .243 |

hierarchical relations (as shown in Section 5.3). Both MuRE and MuRP outperform previous state-of-the-art models on WN18RR on all metrics apart from hits@1, where MuRP obtains second best overall result. In fact, even at relatively low embedding dimensionality ($d = 40$), this is maintained, demonstrating the ability of hyperbolic models to succinctly represent multiple hierarchies. On FB15k-237, MuRE is outperformed only by TuckER [2] (and similarly ComplEx-N3 [18], since Balažević et al. [2] note that the two models perform comparably), primarily due to *multi-task learning* across relations. This is highly advantageous on FB15k-237 due to a large number of relations compared to WN18RR and thus relatively little data per relation in some cases. As the first model to successfully represent multiple relations in hyperbolic space, MuRP does not also set out to include multi-task learning, but we hope to address this in future work. Further experiments on NELL-995, which substantiate our claim on the advantage of embedding hierarchical multi-relational data in hyperbolic over Euclidean space, are presented in Appendix C.

## 5.3 MuRE vs MuRP

**Effect of dimensionality** We compare the MRR achieved by MuRE and MuRP on WN18RR for embeddings of different dimensionalities $d \in \{5, 10, 15, 20, 40, 100, 200\}$. As expected, the difference is greatest at lower embedding dimensionality (see Figure 2a).

**Convergence rate** Figure 2b shows the MRR per epoch for MuRE and MuRP on the WN18RR training and validation sets, showing that MuRP also converges faster.

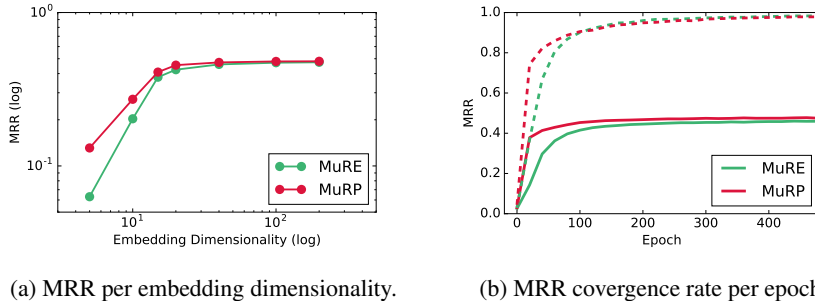

(a) MRR per embedding dimensionality.          (b) MRR covergence rate per epoch.

Figure 2: (a) MRR log-log graph for MuRE and MuRP for different embeddings sizes on WN18RR. (b) Comparison of the MRR convergence rate for MuRE and MuRP on the WN18RR training (dashed line) and validation (solid line) sets with embeddings of size $d = 40$ and learning rate 50.

**Model architecture ablation study** Table 2 shows an ablation study of relation-specific transformations and bias choices. We note that any change to the current model architecture has a negative effect on performance of both MuRE and MuRP. Replacing biases by the (transformed) entity embedding norms leads to a significant reduction in performance of MuRP, in part because norms are constrained to [0, 1), whereas the biases they replace are unbounded.

Table 2: Ablation study of different model architecture choices on WN18RR: relational transformations (left) and biases (right). Current model (top row) outperforms all others.

(a) Relational transformations.

| Distance function | MuRE MRR | MuRE H@1 | MuRP MRR | MuRP H@1 |
|---|---|---|---|---|
| $d(\mathbf{R}\mathbf{e}_s, \mathbf{e}_o + \mathbf{r})$ | **.459** | **.429** | **.477** | **.438** |
| $d(\mathbf{e}_s, \mathbf{e}_o + \mathbf{r})$ | .340 | .235 | .307 | .192 |
| $d(\mathbf{R}\mathbf{e}_s, \mathbf{e}_o)$ | .413 | .381 | .401 | .363 |
| $d(\mathbf{R}_s\mathbf{e}_s, \mathbf{R}_o\mathbf{e}_o + \mathbf{r})$ | .341 | .299 | .367 | .335 |
| $d(\mathbf{e}_s + \mathbf{r}, \mathbf{R}\mathbf{e}_o)$ | .442 | .410 | .454 | .413 |

(b) Biases.

| Bias choice | MuRE MRR | MuRE H@1 | MuRP MRR | MuRP H@1 |
|---|---|---|---|---|
| $b_s$ & $b_o$ | **.459** | **.429** | **.477** | **.438** |
| $b_s$ only | .455 | .414 | .463 | .415 |
| $b_o$ only | .453 | .412 | .460 | .409 |
| $b_x = \|\mathbf{e}_x\|^2$ | .414 | .393 | .414 | .352 |
| $b_x = \|\mathbf{e}_x^{(r)}\|^2$ | .443 | .404 | .434 | .372 |

**Performance per relation** Since not every relation in WN18RR induces a hierarchical structure over the entities, we report the *Krackhardt hierarchy score (Khs)* [17] of the entity graph formed by each relation to obtain a measure of the hierarchy induced. The score is defined only for directed networks and measures the proportion of node pairs $(x, y)$ where there exists a directed path $x \to y$, but not $y \to x$ (see Appendix D for further details). The score takes a value of one for all directed acyclic graphs, and zero for cycles and cliques. We also report the maximum and average shortest path between any two nodes in the graph for hierarchical relations. To gain insight as to which relations benefit most from embedding entities in hyperbolic space, we compare hits@10 per relation of MuRE and MuRP for entity embeddings of low dimensionality ($d = 20$). From Table 3 we see that both models achieve comparable performance on non-hierarchical, symmetric relations with the Krackhardt hierarchy score 0, such as *verb_group*, whereas MuRP generally outperforms MuRE on hierarchical relations. We also see that the difference between the performances of MuRE and MuRP is generally larger for relations that form deeper trees, fitting the hypothesis that hyperbolic space is of most benefit for modelling hierarchical relations.

Computing the Krackhardt hierarchy score for FB15k-237, we find that $80\%$ of the relations have Khs = 1, however, the average of maximum path lengths over those relations is 1.14 with only 2.7% relations having paths longer than 2, meaning that the vast majority of relational sub-graphs consist of directed edges between *pairs of nodes*, rather than trees.

Table 3: Comparison of hits@10 per relation for MuRE and MuRP on WN18RR for $d = 20$.

| Relation Name | MuRE | MuRP | $\Delta$ | Khs | Max Path | Avg Path |
|---|---|---|---|---|---|---|
| also_see | .634 | **.705** | .071 | 0.24 | 44 | 15.2 |
| hypernym | .161 | **.228** | .067 | 0.99 | 18 | 4.5 |
| has_part | .215 | **.282** | .067 | 1 | 13 | 2.2 |
| member_meronym | .272 | **.346** | .074 | 1 | 10 | 3.9 |
| synset_domain_topic_of | .316 | **.430** | .114 | 0.99 | 3 | 1.1 |
| instance_hypernym | **.488** | .471 | −.017 | 1 | 3 | 1.0 |
| member_of_domain_region | .308 | **.347** | .039 | 1 | 2 | 1.0 |
| member_of_domain_usage | .396 | **.417** | .021 | 1 | 2 | 1.0 |
| derivationally_related_form | .954 | **.967** | .013 | 0.04 | – | – |
| similar_to | **1** | **1** | 0 | 0 | – | – |
| verb_group | **.974** | **.974** | 0 | 0 | – | – |

**Biases vs embedding vector norms** We plot the norms versus the biases $b_s$ for MuRP and MuRE in Figure 3. This shows an overall correlation between embedding vector norm and bias (or radius of the sphere of influence) for both MuRE and MuRP. This makes sense intuitively, as the sphere of influence increases to "fill out the space" in regions that are less cluttered, i.e. further from the origin.

**Spatial layout** Figure 4 shows a 40-dimensional subject embedding for the word *asia* and a random subset of 1500 object embeddings for the hierarchical WN18RR relation *has_part*, projected to 2 dimensions so that distances and angles of object entity embeddings *relative to the subject entity embedding* are preserved (see Appendix E for details on the projection method). We show subject and object entity embeddings before and after relation-specific transformation. For both MuRE and MuRP, we see that applying the relation-specific transformation separates true object entities from false ones. However, in the Poincaré model, where distances increase further from the origin, embeddings are moved further towards the boundary of the disk, where, loosely speaking, there is *more space* to separate and therefore distinguish them.

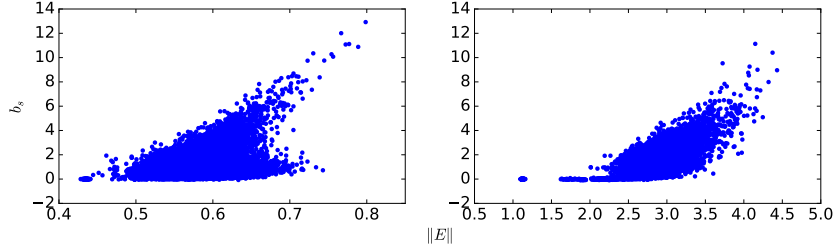

Figure 3: Scatter plot of norms vs biases for MuRP (left) and MuRE (right). Entities with larger embedding vector norms generally have larger biases for both MuRE and MuRP.

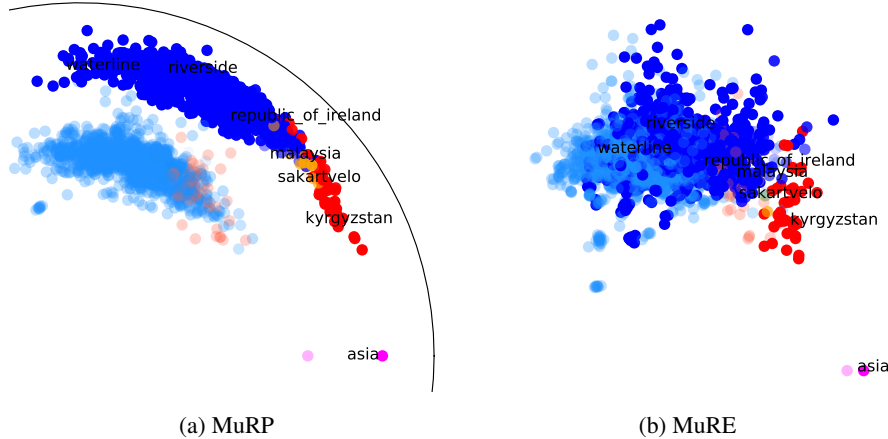

(a) MuRP            (b) MuRE

Figure 4: Learned 40-dimensional MuRP and MuRE embeddings for WN18RR relation *has_part*, projected to 2 dimensions. • indicates the subject entity embedding, • indicates true positive object entities predicted by the model, • true negatives, • false positives and • false negatives. Lightly shaded blue and red points indicate object entity embeddings before applying the relation-specific transformation. The line in the left figure indicates the boundary of the Poincaré disk. The supposed false positives predicted by MuRP are actually true facts missing from the dataset (e.g. *malaysia*).

**Analysis of wrong predictions** Here we analyze the false positives and false negatives predicted by both models. MuRP predicts 15 false positives and 0 false negatives, whereas MuRE predicts only 2 false positives and 1 false negative, so seemingly performs better. However, inspecting the alleged false positives predicted by MuRP, we find they are all countries on the Asian continent (e.g. *sri_lanka*, *palestine*, *malaysia*, *sakartvelo*, *thailand*), so are actually correct, but missing from the dataset. MuRE's predicted false positives (*philippines* and *singapore*) are both also correct but missing, whereas the false negative (*bahrain*) is indeed falsely predicted. We note that this suggests current evaluation methods may be unreliable.

## 6 Conclusion and future work

We introduce a novel, theoretically inspired, translational method for embedding multi-relational graph data in the Poincaré ball model of hyperbolic geometry. Our multi-relational Poincaré model MuRP learns relation-specific parameters to transform entity embeddings by Möbius matrix-vector multiplication and Möbius addition. We show that MuRP outperforms its Euclidean counterpart MuRE and existing models on the link prediction task on the hierarchical WN18RR knowledge graph dataset, and requires far lower dimensionality to achieve comparable performance to its Euclidean analogue. We analyze various properties of the Poincaré model compared to its Euclidean analogue and provide insight through a visualization of the learned embeddings.

Future work may include investigating the impact of recently introduced Riemannian adaptive optimization methods compared to Riemannian SGD. Also, given not all relations in a knowledge graph are hierarchical, we may look into combining the Euclidean and hyperbolic models to produce mixed-curvature embeddings that best fit the curvature of the data.

**Acknowledgements**

We thank Rik Sarkar, Ivan Titov, Jonathan Mallinson, Eryk Kopczyński and the anonymous reviewers for helpful comments. Ivana Balažević and Carl Allen were supported by the Centre for Doctoral Training in Data Science, funded by EPSRC (grant EP/L016427/1) and the University of Edinburgh.

## Footnotes

[1]Each node in a directed graph is connected not only to its children, but to every descendant, i.e. all nodes to which there exists a directed path from the starting node.

[2] https://github.com/ibalazevic/multirelational-poincare

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
