[Supplementary Material]

# A  Poincaré ball model of hyperbolic geometry

The Poincaré ball model is one of five *isometric* models of hyperbolic geometry [7], each offering different perspectives for performing mathematical operations in hyperbolic space. The isometry means there exists a one-to-one distance-preserving mapping from the *metric space* of one model $(\mathcal{X}, d)$ onto that of another $(\mathcal{X}', d')$, where $\mathcal{X}, \mathcal{X}'$ are sets and $d, d'$ distance functions, or *metrics*, providing a notion of equivalence between the models.

Each point on the Poincaré ball $\mathbf{x} \in \mathbb{B}_c^d$ has a *tangent space* $T_{\mathbf{x}}\mathbb{B}_c^d$, a $d$-dimensional vector space, that is a local first-order approximation of the manifold $\mathbb{B}_c^d$ around $\mathbf{x}$, which for the Poincaré ball $\mathbb{B}_c^d$ is a $d$-dimensional Euclidean space, i.e. $T_{\mathbf{x}}\mathbb{B}_c^d = \mathbb{R}^d$. The *exponential map* $\exp_{\mathbf{x}}^c : T_{\mathbf{x}}\mathbb{B}_c^d \to \mathbb{B}_c^d$ allows one to move on the manifold from $\mathbf{x}$ in the direction of a vector $\mathbf{v} \in T_{\mathbf{x}}\mathbb{B}_c^d$, tangential to $\mathbb{B}_c^d$ at $\mathbf{x}$. The inverse is the *logarithmic map* $\log_{\mathbf{x}}^c : \mathbb{B}_c^d \to T_{\mathbf{x}}\mathbb{B}_c^d$. For the Poincaré ball, these are defined [13] as:

$$\exp_{\mathbf{x}}^c(\mathbf{v}) = \mathbf{x} \oplus_c \left( \tanh\left( \sqrt{c}\frac{\lambda_{\mathbf{x}}^c \|\mathbf{v}\|}{2} \right) \frac{\mathbf{v}}{\sqrt{c}\|\mathbf{v}\|} \right) \tag{7}$$

$$\log_{\mathbf{x}}^c(\mathbf{y}) = \frac{2}{\sqrt{c}\lambda_{\mathbf{x}}^c} \tanh^{-1}(\sqrt{c}\| - \mathbf{x} \oplus_c \mathbf{y}\|) \frac{-\mathbf{x} \oplus_c \mathbf{y}}{\| - \mathbf{x} \oplus_c \mathbf{y}\|}. \tag{8}$$

# B  NELL-995-h{100, 75, 50, 25} dataset splits

NELL-995 [36] is a subset of the Never-Ending Language Learner (NELL) [8]. The commonly used test set of NELL-995 [36] contains only 12 out of 200 relations present in the training set, none of which are hierarchical. To ensure a fair representation of all training set relations in the validation and test sets, we create new validation and test set splits by combining the initial validation and test sets with the training set and randomly selecting 10,000 triples each from the combined dataset.

To evaluate the influence of the proportion of hierarchical relations in a dataset on the difference in performance between MuRE and MuRP, we create four subsets of the newly created NELL-995 dataset split, containing 100%, 75%, 50% and 25% hierarchical relations, named NELL-995-h100, NELL-995-h75, NELL-995-h50 and NELL-995-h25, containing 43 hierarchical relations each and 0, 14, 43 and 129 non-hierarchical relations respectively.

# C  NELL-995-h{100, 75, 50, 25} experiments

Table 4 shows link prediction results on the NELL-995-h{100, 75, 50, 25} datasets for MuRE and MuRP at $d = 40$ and $d = 200$. At $d = 40$, MuRP consistently outperforms MuRE on all four datasets. As expected, the difference between model performances gets smaller as we increase the number of non-hierarchical relations and is the smallest on NELL-995-h25. For $d = 200$, MuRE starts to perform comparably (or even outperforms on some metrics) to MuRP on NELL-995-h50 and outperforms on NELL-995-h25. These results substantiate our claim on the advantage of embedding hierarchical data in hyperbolic space, particularly in a scenario where low embedding dimensionality is required.

Table 4: Link prediction results on the NELL-995-h{100, 75, 50, 25} datasets for $d = 40$ and $d = 200$.

| Dataset | Model | $d=40$ | | | | $d=200$ | | | |
| --- | --- | --- | --- | --- | --- | --- | --- | --- | --- |
| | | MRR | Hits@10 | Hits@3 | Hits@1 | MRR | Hits@10 | Hits@3 | Hits@1 |
| NELL-995-h100 | MuRE | .330 | .502 | .366 | .245 | .355 | .527 | .398 | .266 |
| | MuRP | **.344** | **.511** | **.383** | **.261** | **.360** | **.529** | **.401** | **.274** |
| NELL-995-h75 | MuRE | .330 | .497 | .368 | .246 | .356 | **.526** | .396 | .269 |
| | MuRP | **.345** | **.506** | **.382** | **.263** | **.359** | .524 | **.401** | **.275** |
| NELL-995-h50 | MuRE | .342 | .510 | .383 | .256 | **.372** | **.544** | **.415** | **.284** |
| | MuRP | **.356** | **.519** | **.399** | **.271** | .371 | .539 | **.415** | **.284** |
| NELL-995-h25 | MuRE | .337 | .489 | .374 | .259 | **.365** | **.515** | **.404** | **.287** |
| | MuRP | **.343** | **.494** | **.379** | **.266** | .359 | .507 | .397 | .282 |

Figure 5 emphasizes the difference in performance (MRR) of MuRP and MuRE (taken from Table 4). We can see that on the purely hierarchical NELL-995-h100 (43 hierarchical and 0 non-hierarchical

relations), MuRP outperforms MuRE both at lower and higher dimensionality. On the other hand, on NELL-995-h25 which is mostly non-hierarchical (43 hierarchical and 129 non-hierarchical relations), MuRP is only slightly better than MuRE at $d=40$, while MuRE outperforms at $d=200$.

Figure 5: Difference in performance (MRR) of MuRP and MuRE on the NELL-995-h{100, 75, 50, 25} datasets for $d=40$ and $d=200$. The difference becomes smaller (turning negative for $d=200$) as the number of non-hierarchical relations increases.

## D   Krackhardt hierarchy score

Let $\mathbf{R} \in \mathbb{R}^{n \times n}$ be the binary *reachability matrix* of a directed graph $\mathcal{G}$ with $n$ nodes, with $\mathbf{R}_{i,j} = 1$ if there exists a directed path from node $i$ to node $j$ and 0 otherwise. The Krackhardt hierarchy score of $\mathcal{G}$ [17] is defined as:

$$\text{Khs}_{\mathcal{G}} = \frac{\sum_{i=1}^{n} \sum_{j=1}^{n} \mathbb{1}(\mathbf{R}_{i,j} == 1 \wedge \mathbf{R}_{j,i} == 0)}{\sum_{i=1}^{n} \sum_{j=1}^{n} \mathbb{1}(\mathbf{R}_{i,j} == 1)}. \tag{9}$$

## E   Dimensionality reduction method

To project high-dimensional embeddings to 2 dimensions for visualization purposes, we use the following method to compute dimensions $x, y$ for projection $\mathbf{e}_i'$ of entity $\mathbf{e}_i$:

- $e_i^{x\prime} = \frac{\mathbf{e}_s}{\|\mathbf{e}_s\|} \mathbf{e}_i$, $i \in \{s, o_0, o_1, ..., o_N\}$, where $\mathbf{e}_s$ is the original high-dimensional subject entity embedding and $N$ is the number of object entity embeddings.

- $e_i^{y\prime} = \sqrt{\|\mathbf{e}_i\|^2 - \|e_i^{x\prime}\|^2}$, $i \in \{s, o_0, o_1, ..., o_N\}$.

This projects the reference subject entity embedding onto the $x$-axis ($e_s^{x\prime} = \|\mathbf{e}_s\|$, $e_s^{y\prime} = 0$) and all object entity embeddings are positioned relative to it, according to their $e_i^{x\prime}$ component aligned with the subject entity and their "remaining" component $e_i^{y\prime}$.