[Reviews · NeurIPS 2019]

Reviewer 1



The paper is concerned with link prediction in multi-relational datasets which is an important task in AI and ML and fits well into NeurIPS. For this purpose, the paper extends hyperbolic embeddings, which have shown promise on single-relational graphs to the multi-relational setting. As such, the paper makes also a promising contribution to advance hyperbolic representation learning by expanding its use-cases. In general, the proposed method makes sense and is a technically relatively straightforward combination of recent advances in representation learning (e.g., hyperbolic, KG embeddings). However, there are some aspects in the paper that are currently not clear to me: - The introduction seems to argue against hyperbolic embeddings that form the same hierarchy across different relations (which I agree with). However, it seems that the proposed method is doing exactly this by using the same hyperbolic embedding of an entity in all different relations. - Additionally, it is not immediately clear to me why the model makes the specific choice to use both a linear transformation (R) and a ranslation (r) to model different relations. Similarly, what is the motivation for the modeling choice to transform the subject and object embeddings with different functions (diagonal linear transformation vs translation). Did the authors also try conceptually simpler transformations (e.g., only using a full R) and how did they perform? - Regarding the evaluation: The results on WN18RR seem indeed promising. However, the results on FB15k-237 seem to indicate that the proposed model struggles more on classic KGs. It would be interesting to see if this holds up on further KG benchmarks like NELL and YAGO. - It would also be interesting to analyze the the (relative) performance drop of MuRP on FB15k in more detail. While the Krackhardt analysis is informative, it doesn't seem to explain the difference, as directed pairs should also be easy to embed in the Poincare ball. For instance, a possible reason could be that a single embedding hierarchy is likely not a good model for this complex KG (whereas it seems more reasonable for WN18). This analysis could also provide valuable insights into next steps based on the proposed model. - For WN18, I found the Krackhardt analysis to be a valuable contribution which might also get adopted in further papers on hierarchical embeddings. - How does MuRP/E perform when using a comparable embedding to ComplEx-N3 d=2000?

Reviewer 2



This paper defines a new model for knowledge base completion for both euclidean and hyperbolic geometries. Being an experimental paper, I believe the texts spends too much time on the background, and too little on the experiments. Especially since the performances of this model are not far beyond existing performances. Since the model can be discussed and heuristically justified in euclidean space, I would remove most discussions on poincaré space and riemannian optimization as well as the specialization of the model in this space to the appendix. The model defined in euclidean space is s(s, r, o) = -d(R e_s, e_o + r)^2 + b_s + b_o, and is justified as a multi-relational version of s(r, o) = . The heuristic justification is that flexibility on the base space is added by replacing the norms with learned biases. A thorough ablation study for this model is lacking. Regarding the biases : -What are the (euclidean or hyperbolic or both) performances of the model without one bias ? - Without both biases ? - What happens if the biases are not learned but are the non-transformed embedding norms ? - What happens if the biases are the transformed embedding norms ? Regarding the multi-relational transforms : - What happens if we switch matrix transform and translation ? - Why only apply the coordinate rescaling R to one side ? -What happens without one or the other ? Results on all of these would help justify all the elements in this model, which otherwise seems completely arbitrary, beyond some heuristics justifications based on observations on word embeddings. Regarding experiments, authors from [16] made available results as a function of rank that are slightly higher than what is reported in this paper (on par with TuckER). See https://github.com/facebookresearch/kbc for the results. The low ranks performances for MuRP on WN18RR are interesting, and I understand that the use of hyperbolic space is motivated by this. The analysis in "Representation Tradeoffs for Hyperbolic Embeddings" suggests that dimension is not the right metric to compare euclidean and hyperbolic embeddings, since hyperbolic embeddings will make a more thorough use of their floating point representation. An analysis of MurE vs MurP with varying bits / entity (varying both dimensionality and floating points depth) would be very valuable and more convincing. An analysis of the performances of MurE / MurP for higher ranks would also be interesting. As it stands in Figure 2, the performance of the model seems to plateau very quickly with the dimensionality, whereas methods such as [16] reach higher performances for higher ranks. Is the model overfitting beyond this point ? The regularization used is not discussed beyond initalization close to zero. Would an explicit penalty on the embedding norm (or the biases) help ? Or is the discrepancy coming from the learning objective ? (cross-entropy vs 50 sampled negatives and bernoulli neg log-likelihood). Regarding Figure 4, the method for dimensionality reduction seems surprising. The biases are not included in the dimensionality reduction. In this case, why not do a PCA relative to the entity of reference (asia) ? The analysis of this figure seems very anecdotic. If the authors want to justify that current evaluation method under-evaluate the performance difference between MurE and MurP or MurP and other methods, a systematic study should be done, rather than one example. Clarity : the method is clearly described. Significance: the experimental results are not detailed enough to assess the significance of this method. On one dataset, results are on par with the state of the art. On the second they are below the state of the art. Results at low dimensions are mitigated by the propension of hyperbolic methods to use the bits available more fully. The plateau-ing performances for higher dimensions puts in question all the experimental results in this paper and makes it very difficult to assess their significance. Is the limitation coming from the model ? from the loss ? the regularization ? the optimization ? ===================== AFTER REBUTTAL ========================== Thank you for all these new results and precisions, I have no further worries regarding performances for higher dimensions or lower bits of precisions. I am however surprised by the results in the ablation study. Switching R and r seems to lead to a 2 point decrease in MRR for both models. This seems unintuitive, as the effect should be similar to reversing all relations in the dataset (subjects then become objects). Performance of the algorithm should be unaffected by such a reversal. As it stands, I am convinced by the results of the algorithm. However, I don't think the important part of the model or training algorithm that leads to the observed gains have been correctly identified.

Reviewer 3



*After reading the rebuttal, I have increased my score, as the additional ablations, statistical tests, and results on an additional dataset were three key points in my review. Like other reviews, am a bit confused by some aspects of the ablation results (i.e., that reversing R and r makes a difference) and this point should be discussed in the revised version. Originality: An incremental contribution with some insightful methodology - As noted in this paper, there a number of recent works showing how hyperbolic embeddings can be useful (e.g., for embedding ontologies, words, or used more generally in deep learning). This work demonstrates how these ideas can be successfully applied to multi-relational data, using methodologies closely inspired by recent work. Thus, conceptual contribution is useful but not particularly innovative. - While not being exceptionally innovative, the methodology is still insightful---especially the geometric intuition for the bias terms, the use of Mobius addition/multiplication, and the use of the Khs score to interpret the results. Quality: Rigorous and correct, but the investigation could be more thorough - The paper is well-embedded within recent work, includes adequate citations, and the proposed methodology is sound. - The empirical analysis was lacking due to their only being two datasets. Given that the performance trends were very different on these two datasets, it would seem natural to add at least one more dataset to further confirm the observed trends. For example, the Yago ontology (https://www.mpi-inf.mpg.de/departments/databases-and-information-systems/research/yago-naga/yago/) would seem like a good candidate for another "hierarchical dataset" and a biomedical knowledge graph (e.g., http://snap.stanford.edu/biodata/index.html) would seem like a good additional dataset that is less hierarchical. The conclusions in the paper would be greatly strengthened by results on such additional datasets, especially given the differing performance trends on the two datasets that are currently examined. - The results using the Khs score are interesting but they could be made more thorough and rigorous by actually doing some statistical tests regarding the correlation between the Khs score (+ path length) and the performance difference between MuRE and MuRP. Table 2 is interesting, but as a reader I am left wondering how strong, consistent, and stable these trends are. - The bias terms in the score function have a nice geometric intuition, but the other components of the score function---i.e., using a diagonal multiplication on the head entity and translation on the tail---are not as well-motivated. Presumably this score function was chosen after some empirical investigation, and I would expect that this score function performs better than ablations (e.g., only using the DistMult or translational parts). Some more motivation and/or ablation studies regarding the score function would improve the paper. Clarity: The paper is very well-written and well-structured. Significance: Incremental but insightful; useful to a niche community. - The general utility of this approach for learning knowledge graph embeddings is not clear, due to the fact that performance gains depend strongly on the dataset. Given that improvement can only be expected in some datasets, combined with the additional complexities introduced by needing to work in a hyperbolic space, I do not expect this approach to have a significant impact on the general knowledge graph community. Additional results on more datasets and a high-quality code repo could make this work higher impact, however. - This work will certainly be of significant interest to researchers working on hyperbolic embeddings, a relatively niche but growing community.

[Author Response · NeurIPS 2019]

**All reviewers**

**Architecture ablation study:** An ablation study over different model architectures (Table (a)) shows that the chosen
model gives the best performance. Whilst the architecture is in part motivated empirically, it is also based on a recent
theoretical rationale (ICML 2019 Honorable Mention) for using a vector offset (translation) to represent a relation [1].

| R and r ablation | MuRE MRR | MuRE H@1 | MuRP MRR | MuRP H@1 |
|---|---|---|---|---|
| R & r (curr.) | **.459** | **.429** | **.477** | **.438** |
| r only | .340 | .235 | .307 | .192 |
| R only | .413 | .381 | .401 | .363 |
| $R_s$, $R_o$ & r | .341 | .299 | .367 | .335 |
| switch R and r | .442 | .410 | .454 | .413 |

(a) **R/r** ablation study (WN18RR)

| Bias ablation | MuRE MRR | MuRE H@1 | MuRP MRR | MuRP H@1 |
|---|---|---|---|---|
| $b_o$ & $b_s$ (curr.) | **.459** | **.429** | **.477** | **.438** |
| $b_s$ only | .455 | .414 | .463 | .415 |
| $b_o$ only | .453 | .412 | .460 | .409 |
| norms | .414 | .393 | .414 | .352 |
| transf. norms | .443 | .404 | .434 | .372 |

(b) Bias ablation study (WN18RR)

| Model | $d$ | # params | MRR | H@10 | H@3 | H@1 |
|---|---|---|---|---|---|---|
| ComplEx-N3 | 2000 | 160 mil. | .49 | **.58** | .50 | .44 |
| ComplEx-N3 | 500 | 40 mil. | .49 | **.58** | .50 | .44 |
| MuRP | 1000 | 40 mil. | .49 | **.58** | .50 | .44 |
| ComplEx-N3 | 25 | 2 mil. | .44 | .49 | .45 | .41 |
| MuRP | 40 | 1.6 mil | .48 | **.56** | .49 | .44 |

(c) MuRP vs ComplEx-N3 (WN18RR)

4
**Additional datasets:** Comparing performance of MuRP and MuRE ($d = 40$) on NELL-995 (200 relations; 40
hierarchical) shows MuRP outperforms MuRE by ~2%. Looking at relation-specific performance, MuRP outperforms
on hierarchical relations by a larger margin, e.g. "subpartoforganization" by 11%, "specializationof" by 20%.
**Performance on FB15k:** There are two key differences between WN and FB15k datasets: WN is hierarchical with few
relations and many examples per relation; FB15k is non-hierarchical with more relations and less data per relation. WN's
hierarchy favors MuRP. FB15k's lack of hierarchy offers no advantage to hyperbolic embeddings, but its large number
of relations strongly favors multi-task learning (MTL) methods such as TuckER (via core tensor) and ComplEx-N3 (via
rank regularization). Thus, the stronger performance of those methods on FB15k does not show a failure of MuRP, but
highlights the importance of MTL. As the first model to successfully represent multiple relations in hyperbolic space,
MuRP does not also set out to include MTL, but we hope to address this in future work.

We will include all recommendations, e.g. ablation study, statistics and additional experiments, in the paper.

**Reviewer 1**
**Shared entity embeddings:** We share entity embeddings between relations (as in most KBC methods) to learn
relation-agnostic representations of entities that are shared across all relations. These entity embeddings are unlikely
to form a hierarchy with respect to all (if any) relations. Instead they are positioned such that after a relation-specific
transformation, they form a (potentially different) hierarchical structure under each relation of a hierarchical nature.
**Model design, Additional datasets and Performance drop on FB15k:** See "All reviewers".
**ComplEx-N3:** Please note that $d = 2000$ [16] is highly non-standard in the KB literature, where $d = 200$ is the widely
used comparison point. However, we agree that it is important to compare models across a range of dimensionalities.
Table (c) shows MuRP and ComplEx-N3 perform equivalently at $d = 1000$ and $d = 500$ respectively (fair comparison
since ComplEx has imaginary components), and MuRP ($d = 40$) performs comparably even with *25x fewer parameters*.

**Reviewer 2**
**Bias ablation study:** Table (b) shows the impact of changing the biases and that the chosen architecture outperforms
the alternatives considered. Note that for MuRP with biases replaced by (transformed) norms, performance reduces (e.g.
see Hits@1), which is in part because norms are constrained to [0, 1], whereas the biases they replace are unbounded.
**Multi-relational transforms and Justification for architecture:** See "Architecture ablation study".
**Comparison to Facebook repo results:** The results mentioned are not peer-reviewed, so cannot be considered
authoritative, moreover they appeared *after* the submission deadline. However, we include them in Table (c).
**Floating point bits:** The referenced study considers arbitrarily high precision (500+ bits), whereas we use 64 bit
precision across all models for like-for-like comparison. Furthermore, reducing to 32 and 16 bits for MuRE and MuRP
($d = 40$) shows no significant impact, e.g. MRR 0.477 (from 0.477) for MuRP; and 0.457 (from 0.459) for MuRE.
**Performance vs dimension:** The log-log scale of Fig 2a may downplay performance changes at higher dimensionality.
Table (c) shows that performance of MuRP does not plateau at $d = 200$, e.g. with $d = 1000$ (40m params) MuRP performs
similarly to ComplEx-N3, whereas moving to lower dimensionality (~2m params) MuRP shows little performance
drop and outperforms ComplEx-N3, demonstrating the benefit of hyperbolic embeddings at low dimensionality. For
clarity, the results of MuRP/MuRE are achieved without any regularization.
**Figure 4:** The visualization preserves relative distance (whereas PCA would not) between the subject and all object
embeddings. Whilst each object embedding is compared to the subject embedding according to its own bias, we omit
these for clutter but note they are implicitly included by colouring object entities according to their predicted score.

**Reviewer 3**
**Additional datasets and Ablation study:** See "All reviewers".
**Statistical tests:** The Pearson correlation scores (from -1 to 1) between performance difference and (i) '$khs + l$' and
(ii) '$khs \times l$' (for path length $l$) shown in Table 2 are 0.51 and 0.46, respectively, indicating a positive correlation.
**Contribution Significance:** To address the reviewer's doubts about the impact of this study, we note that previous
studies in this line of work have been published in top venues: NeurIPS [5,23,27], ICLR [7,13,28,34], ICML [16,22,32].

[Meta-Review · NeurIPS 2019]

The reviewers appreciated the primary focus of the paper on extending hyperbolic embeddings to multi-relational graphs, and found the proposed approach to be interesting and the results to be fairly strong (on WNET). The reviewers raised a number of concerns that were addressed in the authors' response, however, the additional content, such as the ablation studies and the results on the additional datasets, should be included in the main paper. One remaining concern was the performance when R and r are reversed, which seemed inconsistent with the proposed model (please clarify the results in the revision).